# Development of a Multi-Pass Drawing Process Design System for Steel Profiles

**DOI:** 10.3390/ma11122446

**Published:** 2018-12-03

**Authors:** Sang-Kon Lee, In-Kyu Lee, Sung-Yun Lee

**Affiliations:** Ultimate Fabrication Technology Group, Korea Institute of Industrial Technology, 320, Techno Sunhwan-ro, Yuga-up, Dalseong-gun, Daegu 42994, Korea; lik1025@kitech.re.kr (I.-K.L.); yunskills@kitech.re.kr (S.-Y.L.)

**Keywords:** multi-pass drawing, steel profiles, process design system, FE analysis, drawing experiment

## Abstract

The process design of a multi-pass steel profile drawing depends mainly on the experience of the industrial experts. Therefore, in the actual industrial field, a systematic design method is required to improve the efficiency of process design. Development of such a method will result in a reduction in production cost and time. In this study, a process design system that can be installed in AutoCAD V14 was developed. By using the developed design system, the drawing load, stress, and strain, among other parameters, can be calculated. In addition, intermediate die shapes can be designed. After the process design, both process information and drawings of dies can be obtained. In order to verify the effectiveness of the developed process design system, it was used to design multi-pass profile drawing processes. Finally, finite element (FE) analysis and a drawing experiment were carried out to verify the designed processes.

## 1. Introduction

Multi-pass profile drawing is a typical metal forming process for manufacturing elongated mechanical parts with constant cross-sectional shapes, as shown in Figure 1. The products manufactured by profile drawing have excellent mechanical properties and dimensional accuracy. There are several advantages to this process. First, this process can reduce wastage of material, which consists of up to 13% of the material in the case of profile drawing compared to 80% in the case of conventional machining processes. Second, mechanical properties can be improved owing to the strain hardening effect after drawing. Third, mass production is possible through this process.

In order to produce high-quality products, process design is critical. Until recently, the process design of a multi-pass profile drawing has been mainly dependent on the experience of the designers or experimental techniques and has required a lot of effort. Several studies on multi-pass profile drawing have been carried out. An intermediate die for a multi-pass drawing process has been designed [1]. Yoshida et al. [2] analyzed a three-pass profile drawing of rails for a linear motion guide by using finite element (FE) analysis. In 1988, an expert system that automatically determines the drawing stages was developed by Brücker et al. [3]. Steff et al. [4] compared several design methods for profile drawing and estimated the characteristics of each design method. A utility program for the cold drawing process of hollow sections in order to determine the appropriate tube size, die, mandrel geometries, among other parameters, was developed by Sawamiphakdi et al. [5] in 1998. The influence of factors on the accuracy of drawn rods by using elasto-plasticity equations and the FE method was analyzed by Kampuš [6]. In 1998, in order to improve dimensional accuracy, a new calibration method for complex shape sections with a reflex angle was proposed [7]. Norasethasopn et al. [8] investigated the influences of inclusion shape and size on the drawing of copper shaped wire by using FE analysis. In 2010, the intermediate die shape of a multi-pass profile drawing process was designed by using electric field analysis [9]. Lee et al. [10] studied the manufacturing process of a precision linear guide rail through shape rolling and profile drawing. Lin et al. [11] designed the intermediate die profile of a multi-pass cold drawing process for aluminum wires. However, most of the results obtained in these studies are not easily applicable to the design of a profile drawing process in an actual work site. Although finite element analysis has been widely used, it takes a lot of time and effort. Therefore, more efficient design methods are required.

In a metal forming process, in order to save cost and time, it is very important to design an appropriate process [12,13]. In this study, a process design system that can reduce time and cost was developed for the process design of multi-pass profile drawing. The system can be installed in AutoCAD V14. The graphic user interface (GUI) of the system enables easy operation by the user. By using this system, the intermediate die shape, drawing load, drawing stress, strain, die angle, and reduction area can be determined. Additionally, the drawings of the die can be plotted in AutoCAD through a post-process step. The developed system was applied to design the profile drawing processes for two profiles. Finally, in order to verify the effectiveness of the results, an FE analysis and a profile drawing experiment were carried out.

## 2. Theories for Process Design

### 2.1. Design of an Intermediate Die Shape

In multi-pass profile drawing, the cross-sectional area of the initial material must be greater than that of the final product. The intermediate die cross-sectional shapes are always between the initial material cross-sectional shape and the final product cross-sectional shape. Therefore, in order to design the intermediate die sections, the initial material and the final product cross-sectional shapes are used. As shown in Figure 2, the space between the initial material cross-section and the final product cross-section can be equally divided with a finite number of sections.

Figure 3 describes the method for obtaining the intermediate die sections. The cross-section can be divided into finite sections with a small angle *θ*. In section ①, points *a*, *b*, and points *c*, *d* are the intersection points with the initial material and the final product shape, respectively. As shown in Figure 3, the lines $\bar{ad}$ and $\bar{bc}$ are the section lines between the initial material and the final product. These lines can be divided equally as given in Equation (1):(1-a)$$\bar{b_{1}b_{1,1}}=\bar{b_{1,1}b_{1,2}}=\;\cdots \;=\bar{b_{1,n}c_{1}}=\frac{\bar{b_{1}c_{1}}}{n}$$
(1-b)$$\bar{a_{1}a_{1,1}}=\bar{a_{1,1}a_{1,2}}=\;\cdots \;=\bar{a_{1,n}d_{1}}=\frac{\bar{a_{1}d_{1}}}{n}$$
where *n* is the number of the intermediate sections between the initial material and the final product.

Also, each intermediate section can be obtained as follows:(2)$$\begin{array}{c} S_{1}=\bar{a_{1,1}b_{1,1}}+\cdots +\bar{a_{m,1}b_{m,1}}=\sum_{i=1}^{m}\bar{a_{i,1}b_{i,1}}\\ \vdots \\ S_{n}=\bar{a_{1,n}b_{1,n}}+\cdots +\bar{a_{m,n}b_{m,n}}=\sum_{i=1}^{m}\bar{a_{i,n}b_{m,n}} \end{array}$$
where *S_n_* is the intermediate sections, and *m* is the number of the divided sections of the cross-section as shown in Figure 3. Therefore, the intermediate sections can be obtained by connecting each divided section. Figure 2 shows the intermediate sections between the initial material and the final product when *n* = 4.

In this study, these intermediate sections are used for determining the intermediate die shapes. After the number of intermediate sections is determined by the designer, the outlines of the sections are automatically generated by the design system.

### 2.2. Calculation of the Profile Drawing Load

In the design of the profile drawing process, it is very important to calculate the drawing load. By considering the drawing load, the designer is able to calculate the drawing stress and determine the possibility of drawing. In addition, the specifications of a drawing machine can be determined. It is difficult to calculate the drawing load and stress of the profile drawing owing to the complex shape. Basily et al. [14] calculated the drawing load and stress by using the upper and lower bound solutions. Kim et al. [15] and Lee et al. [16] proposed a new profile drawing load prediction method based on an axisymmetric load prediction model. Trofimov et al. [17] calculated the drawing stress to design the rectangular section. In this study, in order to calculate the drawing load in the process design system, the method proposed by Lee et al. [16] was used.

Figure 4 shows a schematic model of the profile drawing load prediction model. As shown in the figure, the material can be divided into a finite number of small sections.

Figure 5 shows a divided section of the profile drawing material. As shown in the figure, it can be assumed that the divided section undergoes an axisymmetric drawing process in which the initial radius and the final radius are *R*_(*i*)_ and *R*_(*e*)_, respectively. Therefore, the drawing load of the divided section can be predicted by Equation (3):(3)$$L=AL\times \frac{\theta }{2\pi }$$
where *L* is the drawing load of the divided section, *AL* is the drawing load for axisymmetric drawing, and *θ* is the central angle of the divided section.

The total profile drawing, using Equation (3), is expressed as:(4)$$L_{total}=\underset{i=n}{\sum }L_{i}$$
where *L_total_* is the total profile drawing load, and *n* is the number of divided sections.

From Equation (3), it can be seen that in order to calculate the profile drawing load, an axisymmetric load prediction model is required. Several axisymmetric load prediction models have been proposed. This study used the axisymmetric load prediction model based on the upper boundary method proposed by Geleji [18]. It can be expressed as follows:(5)$$AL=k_{m}\left(F+Q\cdot \mu \right)+0.77\cdot k_{fm}\cdot A_{2}\cdot \alpha$$
where *k_m_* is the average deformation resistance of the material; *F* is the difference between the cross-sectional areas of the material at the inlet and the exit of the die; *Q* is the contact area between the material and the die; *μ* is the friction coefficient between the material and the die; *k_fm_* is the mean yield strength of the before and after deformation of the material; *A*_2_ is the cross-sectional area of the exit of the die; and *α* is the semi-die angle.

Therefore, from Equations (3)–(5), the following equation for the total profile drawing load can be obtained:(6)$$L_{total}=\underset{i=n}{\sum }L_{i}=\underset{i=n}{\sum }\left[\left\{k_{m}\left(F_{i}+Q_{i}\cdot \mu_{i}\right)+0.77\cdot k_{fm,i}\cdot A_{2,i}\cdot \alpha_{i}\right\}\times \frac{\theta_{i}}{2\pi }\right]$$

### 2.3. Determination of the Number of Passes

The determination of the number of passes is very important. In general, two or three passes are required for profile drawing. In the previous study, the possible total reduction in the cross-sectional area could range from 10–40% per pass [5]. In this study, in order to determine the number of passes, the average reduction area and the drawing stress were used.

The average reduction in area (*r_avg_*) can be expressed as follows [19]:(7)$$r_{avg}=\left[1-{\left(1-r_{total}/100\right)}^{1/n}\right]\times 100$$
where *r_total_* is the total reduction area from the initial material to the final product, and *n* is the number of passes.

Furthermore, from Equation (6), the drawing stress can be expressed as follows:(8)σ=LtotalA2

## 3. Process Design System

### 3.1. Procedure of Process Design

Based on the above theories, a process design system was developed. The design procedure of the system is shown in Figure 6 and is described below.

Step 1. Recognition of the final product shape drawn in AutoCAD

In this step, the area, perimeter, shape factor, coordination of the centroid, and diameters of the minimum and maximum circumscribed circles of the final product shape are calculated. These values are the basic data for the process design.

Step 2. Classification of profile drawing

In this step, the profile drawing process is classified. Most profiles can be manufactured using only the profile drawing process. However, special profiles need to be pre-rolled. In this study, the criteria of process classification were established. Figure 7 shows the procedure of classification.

In Figure 7, *SP* is the number of planes of symmetry of the final shape. As the number of planes of symmetry of a circle is infinite, an increase in the number of planes of symmetry signifies a simpler final shape. *MMR* is the radius ratio and is expressed as follows:(9)MMR=RmaxRmin
where *R_max_* and *R_min_* are the radii of the maximum and minimum circumscribed circles of the final product shape, respectively. The decrease in *MMR* also signifies a simpler final shape.

*SF* is the shape factor and is written as follows:(10)SF=P2A2
where *P* is the perimeter of the final shape. A higher *SF* value signifies a more complex cross-section of the final product.

Step 3. Selection of material

In this step, the user selects the material from the material database in order to calculate the drawing load and stress. The properties of each material listed in the database are obtained from tensile test data.

Step 4. Creation of intermediate sections

In this step, the intermediate sections between the initial material cross-sectional shape and the final product cross-sectional shape are determined based on the input dividing number, as shown in Figure 3.

Step 5. Selection of the intermediate die shape

In this step, the appropriate intermediate die shapes are determined based on the reduction in the cross-sectional area and the intermediate sections from Step 4, as well as the drawing stress and the designer’s experience.

Step 6. Process analysis

In this step, the designed profile drawing process is analyzed. The reduction in cross-sectional area, the drawing load, the drawing stress, and the die half angle are calculated. After this step, if the user wishes to modify the die shape, steps 4 to 6 are repeated.

### 3.2. Design Assist Program

Based on the above theories, the process design assist program was developed. As shown in Figure 8, the design program was coded using Visual C# with .Net. The developed program can be installed on AutoCAD V14.

Figure 9 shows the main GUI window of the developed system installed in AutoCAD V14. As shown in the figure, the main window is composed of nine sections: ① final product section, ② process analysis section, ③ material information section, ④ virtual die section, ⑤ die information section, ⑥ die update section, ⑦ drawing die information section, ⑧ detailed information section, ⑨ post-process section.

## 4. Application of the Process Design System

To verify the effectiveness of the developed process design system, a profile drawing process design was carried out for two profiles: one with a diamond-shaped cross-section, the other with a teardrop-shaped cross-section. Figure 10 shows the cross-sections of the two profiles.

The areas of the diamond and teardrop cross-sections were 1319.9 mm^2^ and 333.9 mm^2^, respectively. The conditions for the process design are shown in Table 1.

The flow stress of AISI 1045 steel is as follows:(11)$$\bar{\sigma }=836{\bar{\epsilon }}^{0.22}\;\left(\mathrm{MPa}\right)$$

Based on the diameter of the minimum circumscribed circle of the profiles and the designer’s experience, the diameters of the initial materials for the diamond and teardrop cross-sections were set at 50.0 mm and 26.4 mm, respectively. Therefore, the total reductions in the cross-sectional areas of each profile were 32.78% and 39.0%, respectively. Generally, the maximum reduction in the cross-sectional area of one pass was 30.0%. Therefore, it can be determined that a two-pass profile drawing is required to produce the applied two profiles. From Equation (7), the average reductions in the cross-sectional areas of the two-pass drawing processes were 18.0% and 21.9%, respectively.

The results of the process analysis are shown in Figure 11. From these results, it can be determined that the two profiles can be produced through profile drawing without pre-rolling.

The results of the process design are shown in Figure 12. After considering these results, the sixth sections were selected for the first passes. In this result, the drawing stresses at the eighth and seventh sections exceeded the yield strength of the drawing material. This means that it is impossible to manufacture the final products through a one-pass drawing. Therefore, a two-pass profile drawing is applied.

The intermediate sections and the selected first pass sections are shown in Figure 13.

Table 2 shows the reduction areas with each pass. The reduction area from the first pass was higher than that from the second pass.

The designed profile drawing processes plotted in AutoCAD for the diamond and teardrop cross-sections are presented in Figure 14 and Figure 15, respectively. As mentioned above, the cross-sectional shape, the deformed shape of the material, and the drawings of the dies can be obtained.

## 5. Numerical and Experimental Validation

### 5.1. Numerical Validation of the Designed Process

To validate the designed drawing processes, FE analysis was carried out using DEFORM 3D software. Figure 16 shows the initial FE analysis models for the designed drawing processes. A half model was applied considering the plane of symmetry. A list of the FE analysis conditions is provided in Table 3.

Figure 17 shows the distribution of the effective strain after the first pass. The effective strain was higher on the surface of the material than inside the material because of higher deformation on the surface. The effective strains on the surface after the first pass for the diamond and teardrop cross-sections were 0.310–0.387 and 0.325–0.415, respectively.

Figure 18 shows the results of the second pass. Before the second drawing, the effective strain was set to zero to take into account the annealing heat treatment after the first pass. The heat treatment was applied to improve the formability of the material after the first pass. The effective strains on the surface of the diamond and teardrop sections after the second pass were 0.308–0.365 and 0.215–0.335, respectively.

Figure 19 shows the drawing loads obtained from FE analysis. In both cases, the drawing load was higher for the first pass than the second pass because the material size was larger for the first pass than the second pass.

### 5.2. Profile Drawing Experiment

Finally, a profile drawing experiment was conducted to verify the results of the FE analysis and the developed process design system. Figure 20 shows the profiles of the drawing dies used in the experiment.

Figure 21 shows the drawn final products after the profile drawing experiment. The results of the experiment show that the dimensional tolerances of the final products were within the specified tolerance of ±0.05 mm.

Table 4 shows a comparison of the drawing loads obtained by the design system, the FE analysis, and the experiment. The drawing loads were consistent and the maximum difference between the results was approximately 11%. Therefore, it can be concluded that the developed design assist program can be effectively applied to the process design of multi-pass profile drawing.

## 6. Conclusions

To produce high-quality steel profiles, it is essential to design an appropriate profile drawing process. In this study, a process design system was developed to design a multi-pass profile drawing process. The developed design system can be installed on AutoCAD V14 and allows easy operation by the user. The following conclusions were obtained from an investigation of this program.

(1)In this study, the criteria of process classification were established. It can be determined whether the pre-rolling process is necessary or not based on the criteria.(2)The intermediate die shapes for multi-pass profile drawing were determined by using the intermediate sections between the initial and final cross-sectional shapes, the reduction area, the yield strength of the material, and the designer’s experience.(3)The developed process design system is operated through a user-friendly GUI window. This window provides the user with diverse information including the final shape, the reduction area, the drawing load, the drawing stress, and the strain. Based on this information, as well as the designer’s experience, the process design can be carried out.(4)To verify the effectiveness of the design system, multi-pass profile drawing processes for two profiles (one with a diamond cross-sectional shape the other with a teardrop cross-sectional shape) were designed. The results of the design system, FE analysis, and experiment showed that the drawing loads obtained were consistent with a maximum difference between drawing loads of approximately 11%. Moreover, the final profile obtained by experiment had a dimensional accuracy within the specified tolerance.(5)The results of this study show that the developed design system is a useful method for designing the multi-pass profile drawing process. Future studies need to build on this study by providing an additional capability of designing the pre-rolling process. This will allow the design system to be used more effectively.

## Figures and Tables

**Figure 1 materials-11-02446-f001:**
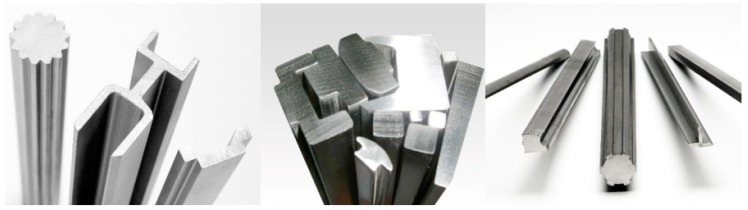
Profiles manufactured by profile drawing process.

**Figure 2 materials-11-02446-f002:**
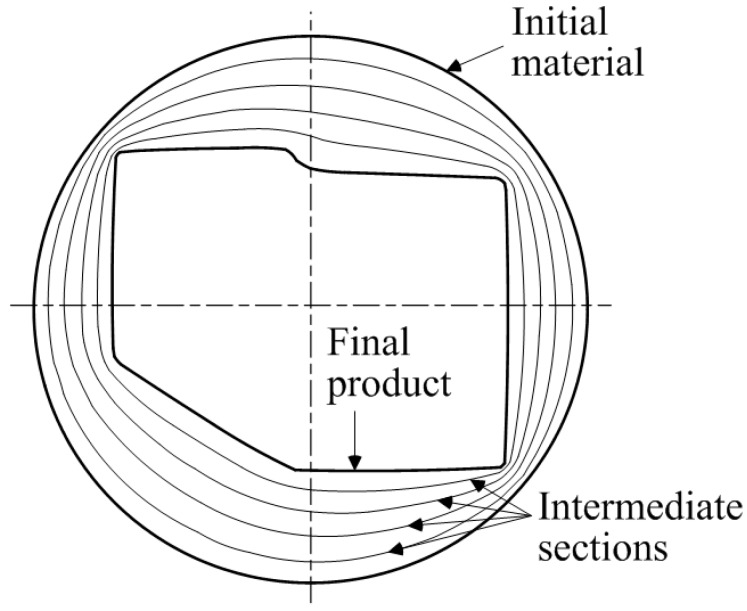
Intermediate sections between the initial and final cross-sections.

**Figure 3 materials-11-02446-f003:**
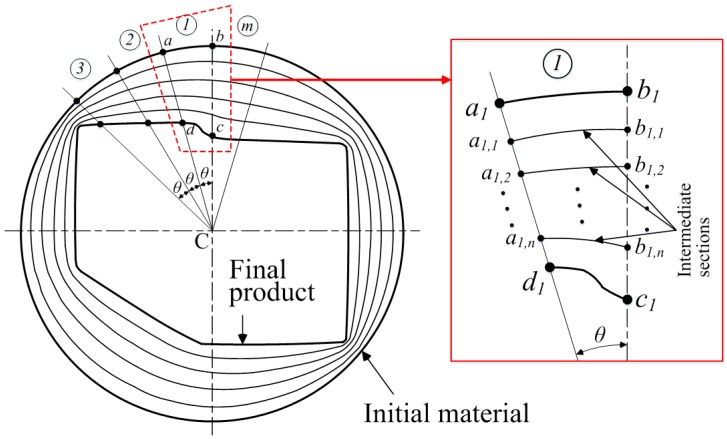
Method for obtaining the intermediate sections.

**Figure 4 materials-11-02446-f004:**
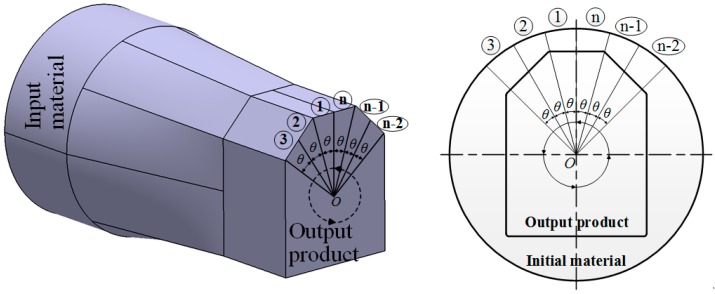
Divided small sections.

**Figure 5 materials-11-02446-f005:**
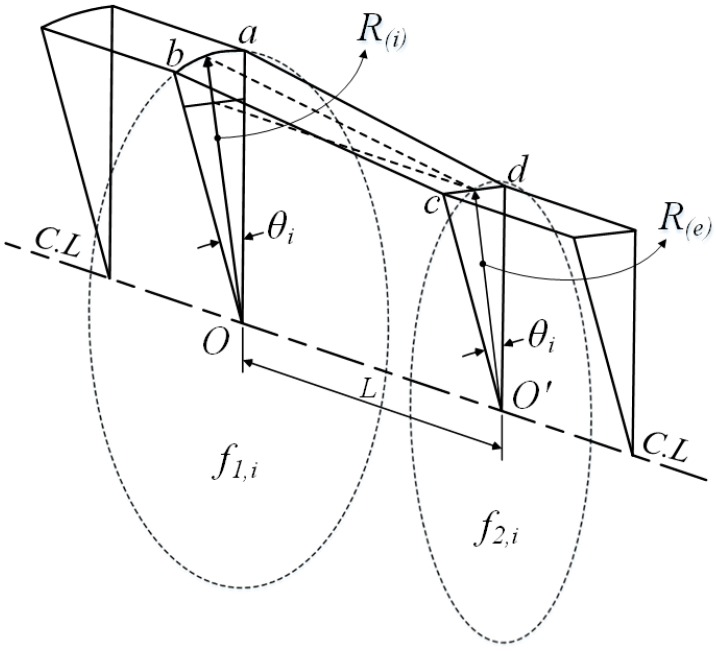
Iso-view of a divided section.

**Figure 6 materials-11-02446-f006:**
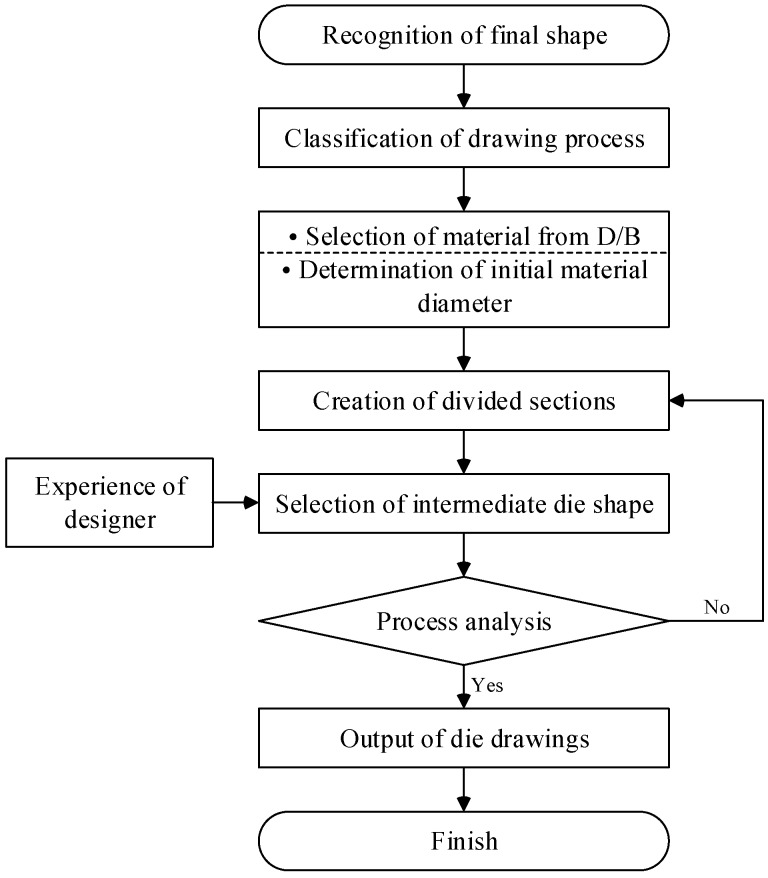
Design procedure of the profile drawing process.

**Figure 7 materials-11-02446-f007:**
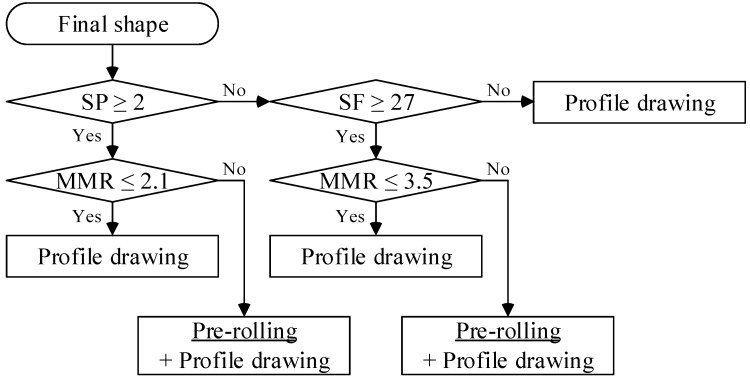
Procedure for classification of profile drawing.

**Figure 8 materials-11-02446-f008:**
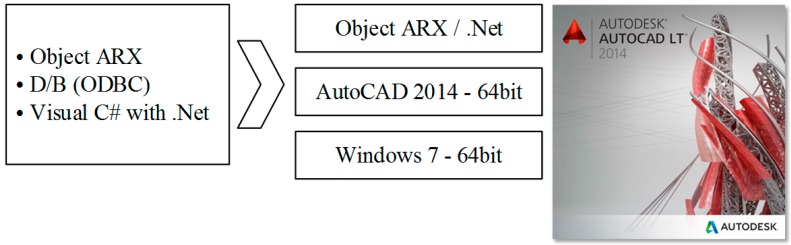
Developing the environment of the process design system.

**Figure 9 materials-11-02446-f009:**
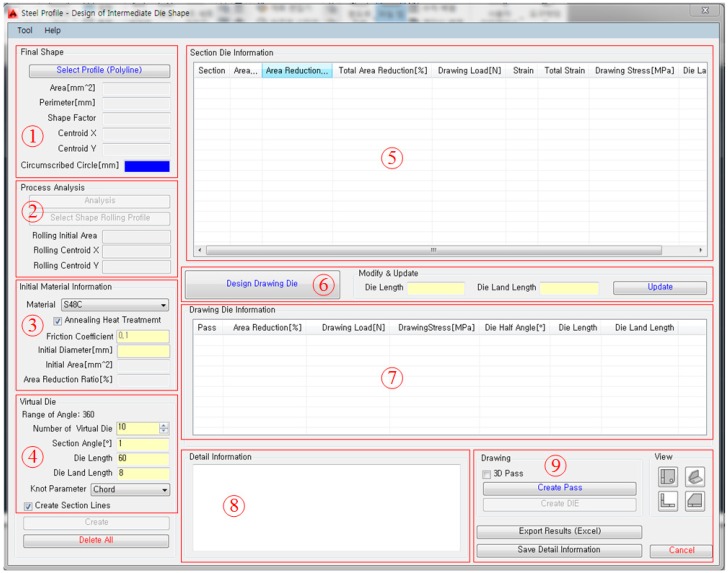
Main graphic user interface (GUI) window of the developed design assist system.

**Figure 10 materials-11-02446-f010:**
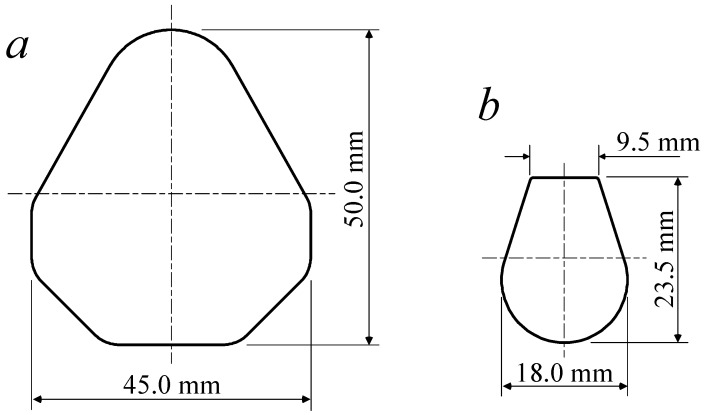
Applied profiles. (**a**) Diamond cross-section; (**b**) teardrop cross-section.

**Figure 11 materials-11-02446-f011:**
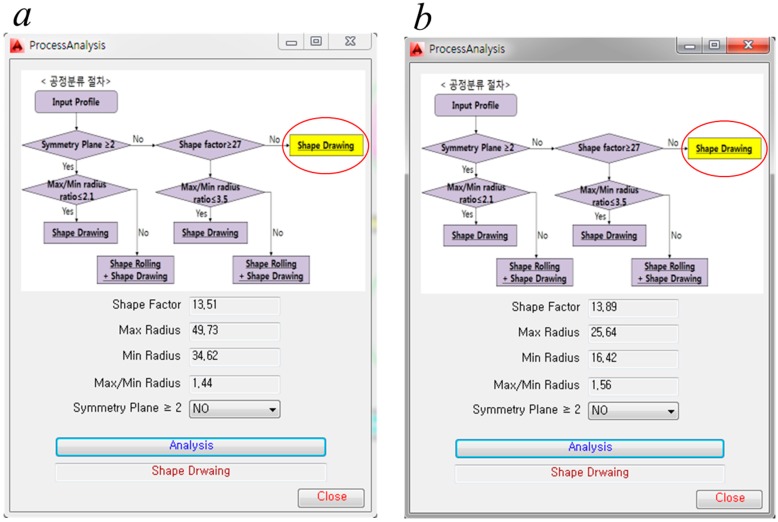
Results of the process analysis. (**a**) Diamond cross-section; (**b**) teardrop cross-section.

**Figure 12 materials-11-02446-f012:**
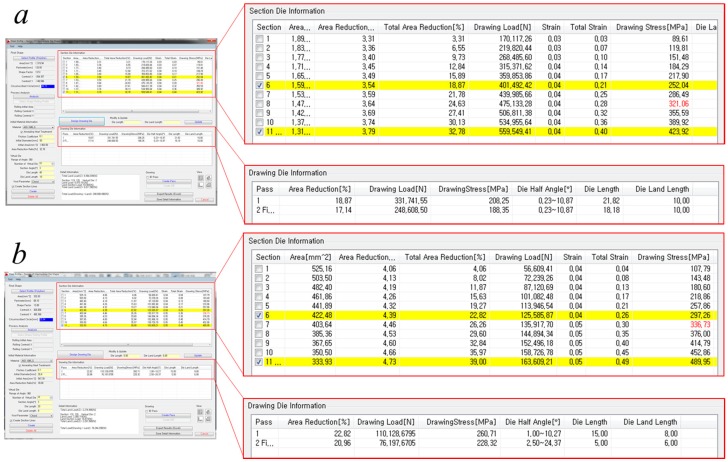
Process design. (**a**) Diamond cross-section; (**b**) teardrop cross-section.

**Figure 13 materials-11-02446-f013:**
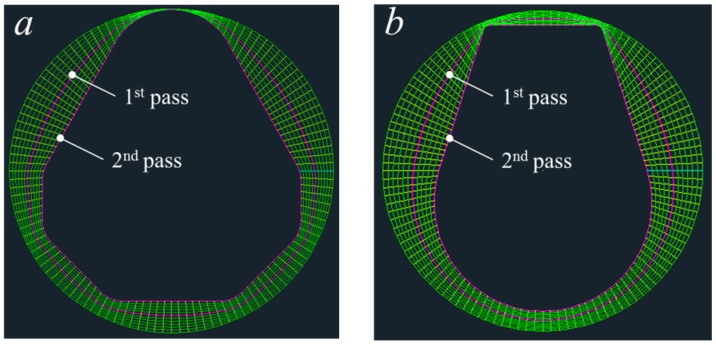
Intermediate sections. (**a**) Diamond cross-section; (**b**) teardrop cross-section.

**Figure 14 materials-11-02446-f014:**
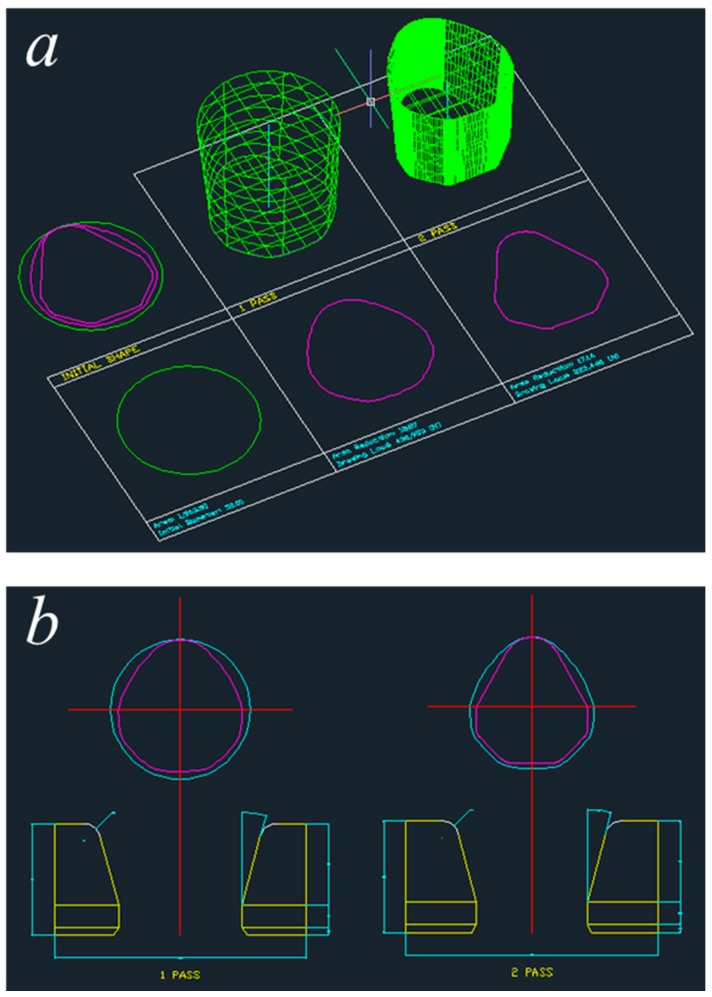
Designed process for the diamond cross-section. (**a**) Pass information; (**b**) drawings of the die.

**Figure 15 materials-11-02446-f015:**
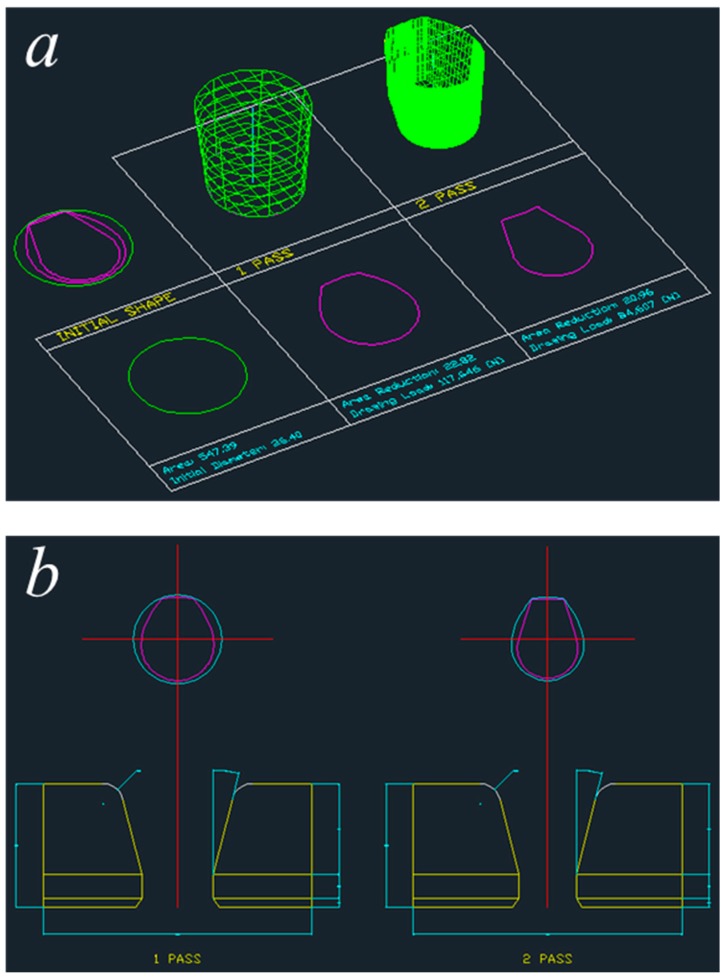
Designed process for the teardrop cross-section. (**a**) Pass information; (**b**) drawings of the die.

**Figure 16 materials-11-02446-f016:**
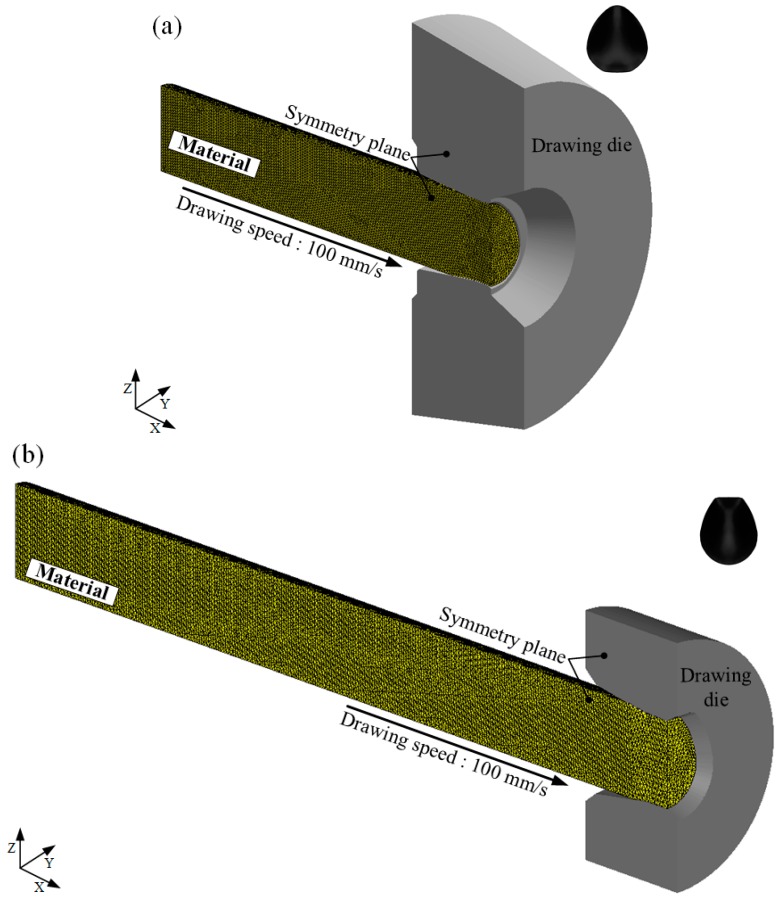
Initial finite element (FE) analysis models. (**a**) Diamond cross-section; (**b**) teardrop cross-section.

**Figure 17 materials-11-02446-f017:**
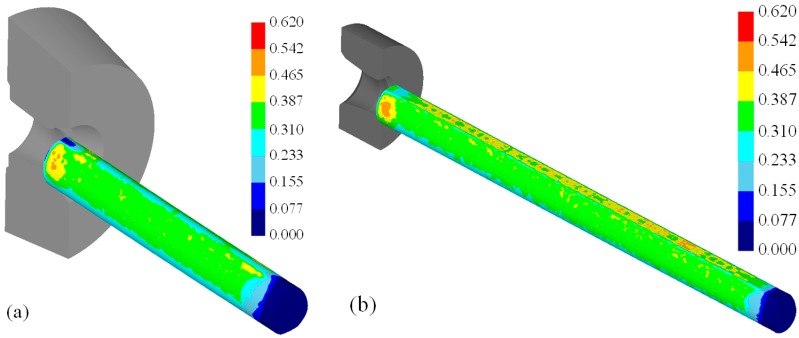
Distribution of the effective strain after the first pass. (**a**) Diamond cross-section; (**b**) teardrop cross-section.

**Figure 18 materials-11-02446-f018:**
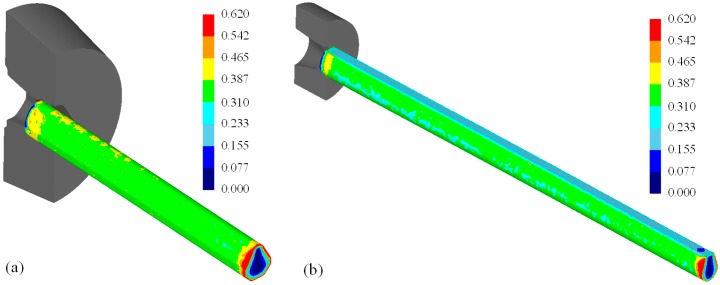
Distribution of the effective strain after the second pass. (**a**) Diamond cross-section; (**b**) teardrop cross-section.

**Figure 19 materials-11-02446-f019:**
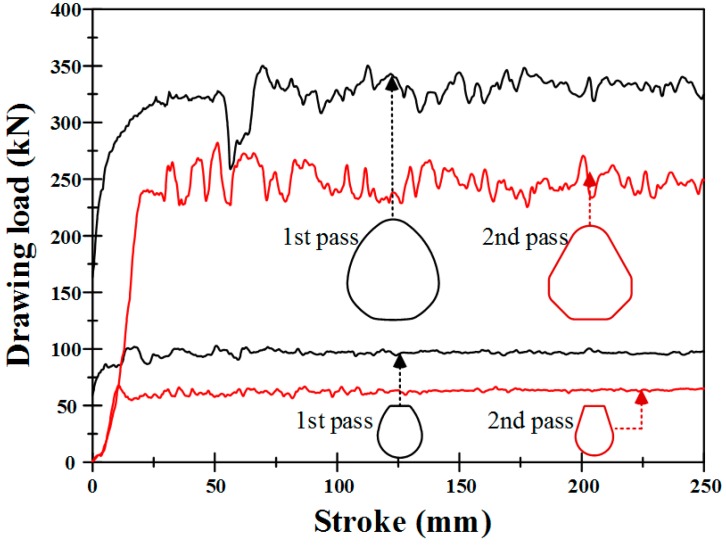
Drawing loads from the results of FE analysis.

**Figure 20 materials-11-02446-f020:**
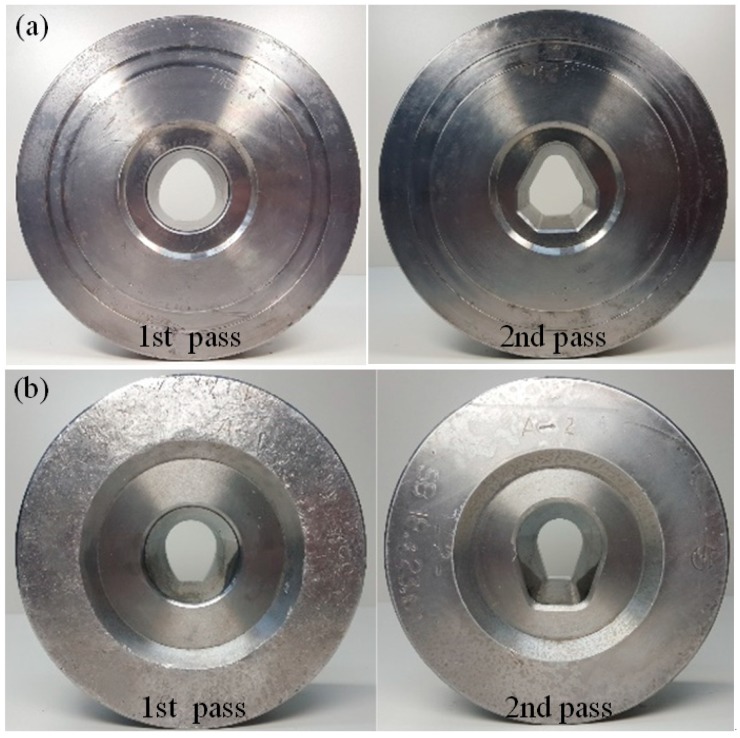
Profiles of the drawing die. (**a**) Diamond cross-section; (**b**) teardrop cross-section.

**Figure 21 materials-11-02446-f021:**
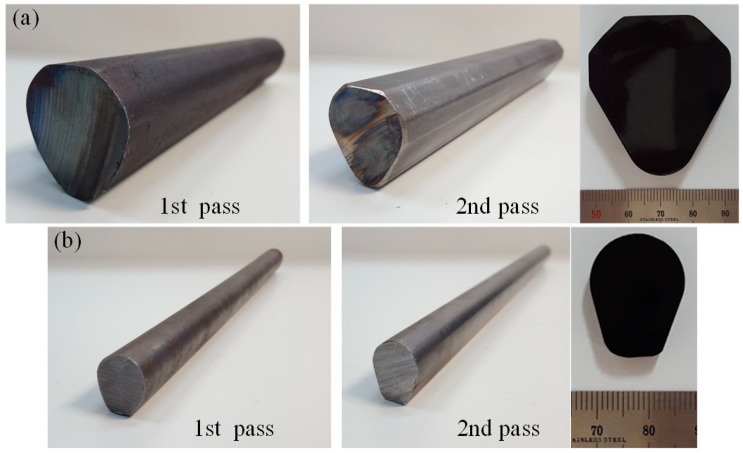
Drawn final products. (**a**) Diamond cross-section; (**b**) teardrop cross-section.

**Table 1 materials-11-02446-t001:** Conditions for the profile drawing process design.

Conditions	Diamond Cross Section	Teardrop Cross Section
Area of the profile (mm^2^)	1319.94	333.93
Perimeter of the profile (mm)	133.55	68.10
Shape factor	13.51	13.89
Diameter of the minimum circumscribed circle (mm)	49.73	25.64
Material	AISI 1045	AISI 1045
Annealing heat treatment between passes	apply	apply
Friction coefficient (μ)	0.05	0.05
No. of intermediate sections	10	10
Die length between inlet and exit of die (mm)	40.0	20.0
Die land length (mm)	10	8

**Table 2 materials-11-02446-t002:** Reduction areas of each pass.

Profile	First Pass	Second Pass
Diamond cross section	18.8%	17.1%
Teardrop cross section	22.8%	20.9%

**Table 3 materials-11-02446-t003:** Finite element (FE) analysis conditions.

Conditions	Diamond Cross-Section	Teardrop Cross-Section
Drawing velocity (mm/s)	100.0	100.0
No. of elements	137,825	196,177
No. of nodes	29,911	41,803
Friction coefficient (*μ*)	0.05	0.05

**Table 4 materials-11-02446-t004:** Comparison of drawing loads (kN).

Method	Pass	Diamond Cross-Section	Teardrop Cross-Section
FE analysis	1	318.8	101.0
	2	237.4	65.7
Design program	1	331.6	109.9
	2	248.2	73.6
Experiment	1	316.9	103.0
	2	240.3	69.7

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
