# Peer review of "Development of a Multi-Pass Drawing Process Design System for Steel Profiles"

_materials, 2018, doi:10.3390/ma11122446_

Round 1
Reviewer 1 Report
The subject matter of the paper is up-to-date and interesting. For the purpose of clarity, the following revisions should be made to the manuscript:
The equation (9) is given wrong, because: MMR=Rmax/Rmin (in the paper is MMR=Rmin/Rmax)
The figure 15a should be improved, because diamond cross section has a bad proportions. In the respect to dimensions the height and the width of the section are equal (on the figure height is greater than the width).
The drawing loads calculated by FEM and listed at table 4 are 10 times less than presented on figure 24. Which data are proper?
Author Response
Thank you for your kind comment.
I can see the detailed answers in the attached file.

Reviewer 2 Report
Overall:
The paper does not introduce new scientific value. At least, it does not become clear. The authors developed a user interface that uses existing models. Questions that would include scientific value, as on optimizing the intermediate section numbers and shapes are not discussed and still to be decided upon by the user.
The paper introduces an interesting piece of UI programming that could be worth being published. It does not contribute new scientific knowledge. It seems to be rather a report than a scientific paper. This has to be made clear in the text.
The following hints are just a selection. The whole paper has to be revised in that sense.
lines 10, 11
This needs to be proven. I am sure, industry has very sophisticated and systematic tools that might even be better than the tool suggested in this paper. Therefore, the motivation has to be reformulated.
lines 20ff
the literature review does a lot of name-dropping but does not explain scientific reasons behind the development of this UI.
Eq. 5 / Fig. 11
Which material parameters were used? The parameters in Eq. 5: km and kfm do not match the parameters in Fig. 11. Are km and kfm calculated form the values in Fig. 11? How? What would be the influence eg. of changes in strain hardening?
Fig. 6
Contrary to the introduction to their paper, the author now require the "experience of designer" for the most important input to their process.
line 169
The method only includes a decision on the number of intermediate sections. The design of each section shape once the number of shapes is fixed, will influence the result very much. A discussion on the section shapes is necessary.
section 5: validation
Since there is no new model developed, what exactly is to be validated? The preceding sections mainly focus on the drawing force. The validation focusses on strain. Why? Even the description of the experiment does not focus on the drawing force.
Figure 24:
Why is a FE calculation strongly fluctuating? Why are the experimental results and the UI calculation values not shown in this figure?
Author Response

(The authors gave the same response as above.)

Reviewer 3 Report
In the reviewer’s opinion, this work features some contribution towards the enhancement in the efficiency of design process. Overall, the paper is generally properly written, however there are some minor remarks, which if taken into account by the authors, may improve the quality of the manuscript:
Please notice – pXXlYY: page XX line YY of manuscript
1. p14l247: The hardening exp = -0.22 vs. fig. 11 strain hardening exp. = 0.23. The differences are also notice for strain hardening coefficient. Please correct.
2. In fig.11 the yield strength for AISI1045 was defined as 500 MPa so according to fig. 17 one pass drawing should be possible. Why did the authors state different in p14l260? Please clarify.
3. p22l341: The first conclusion is not supported by any research – it is rather an "ad hoc" criterion, as its correctness is not proved. Several experiments should be concluded to check if SP=2 SP=27 are critical values. The critical values of MMR parameters should be also supported by FEM approach or state-of-the-art.
4. In addition, such a detailed presentation of the designed system seems not to be the most interesting for possible readers. The outcomes may be implemented in many different CAD systems. The reviewer would recommend shortening of the detailed description of program functionality (e.g.: fig. 9 and the description).
Author Response
Thank you for your kind comment.
You can see the detailed answers in the attached file.
